# Indocyanine Green and Methyl-β-Cyclodextrin Complex for Enhanced Photothermal Cancer Therapy

**DOI:** 10.3390/biomedicines8110476

**Published:** 2020-11-05

**Authors:** Gayoung Jo, Bo Young Lee, Eun Jeong Kim, Min Ho Park, Hoon Hyun

**Affiliations:** 1Department of Biomedical Sciences, Chonnam National University Medical School, Hwasun 58128, Korea; jky6213@naver.com (G.J.); 0000by@naver.com (B.Y.L.); 2Department of Surgery, Chonnam National University Medical School, Hwasun 58128, Korea; angeleunei@naver.com

**Keywords:** photothermal therapy, indocyanine green, methyl-β-cyclodextrin, inclusion complex, near-infrared fluorescence imaging

## Abstract

A feasible and biocompatible supramolecular complex self-assembled from indocyanine green (ICG) and methyl-β-cyclodextrin (Mβ-CD) was developed for targeted cancer imaging, which enhanced fluorescence-guided photothermal cancer therapy. This study confirmed that the formation of an inclusion complex of the heterocyclic ICG moiety and Mβ-CD inner cavity could result in improved tumor targetability compared with free ICG. The ICG-CD complex could be used as a bifunctional phototherapeutic agent for targeted cancer phototherapy due to the high tumor targetability of the Mβ-CD moiety and effective photothermal performance of the near-infrared (NIR) ICG moiety. Upon NIR laser irradiation, the photothermal effect exerted by the ICG-CD complex significantly enhanced the temperature at the tumor site by 56.2 °C within 5 min. Targeting HT-29 tumors using the ICG-CD complex resulted in an apparent reduction in tumor volumes over the 9 days after photothermal treatment. Moreover, no tumor recurrence or body weight loss were observed after administering a single dose of ICG-CD complex with NIR laser irradiation. Therefore, the administration of the biocompatible ICG-CD complex in combination with NIR laser treatment can be safely explored as a potential strategy for future clinical applications.

## 1. Introduction

Cancer phototherapy associated with a phototheranostic agent has garnered considerable interest in recent years, owing to the feasibility of the approach for improving therapeutic effects [1,2,3]. An ideal phototheranostic agent is required to exhibit important characteristics including biocompatibility, imaging ability, tumor targetability, and phototherapeutic ability [4,5,6,7,8]. Among the numerous phototherapeutic agents, indocyanine green (ICG) is a clinically available near-infrared (NIR) fluorophore that serves multiple roles as a photothermal agent, photosensitizer, and fluorescence imaging probe [9,10,11]. However, the use of ICG alone is not ideal because it is unstable in aqueous solutions and does not exhibit considerable accumulation at the tumor for effective NIR image-guided photothermal therapy (PTT) [12,13]. Although the incorporation of ICG into various nanomaterials is a popular method to improve its accumulation at the tumor site for targeted cancer phototherapy, the use of nanomaterials combined with ICG are still constrained by practical problems, such as complicated synthetic processes and unresolved biosafety issues [14,15,16,17,18]. Therefore, a novel strategy to improve the tumor targetability of ICG is required to enhance its application in photothermal cancer therapy.

Since the biocompatibility of a molecule is an important consideration for its potential clinical application, an alternative method is to protect ICG from its aqueous environment through inclusion complexation with non-toxic cyclodextrin (CD) molecules. CDs, which belong to a family of cyclic oligosaccharides with a hydrophobic interior and hydrophilic exterior, are widely used in pharmaceutics to improve the water solubility, stability, and bioavailability of hydrophobic molecules [19,20]. Barros et al. have previously reported the binding between ICG and β-CD in aqueous solution and identified in terms of aggregation and complexation equilibria of this process [21].

Methyl-β-cyclodextrin (Mβ-CD), a CD derivative, is more water-soluble than β-CD, and it has been used in numerous previous studies to remove plasma membrane cholesterol due to its higher cholesterol affinity than that of β-CD [22,23]. In addition, Grosse et al. have reported that Mβ-CD shows significant antitumor activity in human tumor xenograft models [24]. However, the combination of Mβ-CD and ICG forming an inclusion complex for NIR fluorescence-guided PTT upon single laser irradiation has not been reported. Therefore, developing a rational strategy to assemble ICG and Mβ-CD assumes significance in the development of a targeted phototherapeutic agent to enhance the precision of tumor diagnosis and image-guided PTT.

In the present study, we prepared a tumor-targeted phototheranostic ICG-CD complex rationally assembled from ICG and Mβ-CD in an aqueous state for enhanced photothermal cancer therapy. Importantly, Mβ-CD played a key role as a protective cap to prevent aggregation and improve water solubility and tumor targetability of ICG. Consequently, the ICG-CD complex showed markedly higher accumulation at the tumor site than free ICG. Upon 808 nm NIR laser irradiation, effective PTT with notable suppression of tumor growth was observed in a HT-29 tumor-bearing mouse model following treatment with the ICG-CD complex. Collectively, our results show that the biocompatible ICG-CD complex combined with NIR laser treatment can be safely used as a promising strategy for future clinical applications.

## 2. Experimental Section

### 2.1. Preparation of ICG-CD Complex

ICG and Mβ-CD were purchased from Sigma-Aldrich (St. Louis, MO, USA) and used as received without further purification. The ICG-CD complex was prepared using a host–guest self-assembly process. Briefly, 1 mg (1.3 µmol) of ICG and 5 mg (3.8 µmol) of Mβ-CD were completely dissolved in 0.5 mL methanol and 1.5 mL deionized (DI) water, respectively. Next, the ICG solution was added into the Mβ-CD aqueous solution dropwise to obtain the ICG-CD complex, followed by over 6 h of stirring. The mixture was further dialyzed against DI water using a 2 kDa molecular weight cut-off membrane for 3 days to remove free ICG, Mβ-CD monomers, and residual methanol. Subsequently, the dialyzed solution was lyophilized and dispersed in phosphate-buffered saline (PBS, pH 7.4), especially important when performing in vivo experiments.

### 2.2. Analysis of Optical Properties

All optical measurements were performed at 37 °C in PBS at pH 7.4. Absorption spectra of free ICG and the ICG-CD complex were measured using a fiber optic FLAME spectrometer (Ocean Optics, Dunedin, FL, USA). The fluorescence emission spectra of free ICG and the ICG-CD complex were analyzed using a SPARK^®^ 10M microplate reader (Tecan, Männedorf, Switzerland) at an excitation wavelength of 730 nm and emission wavelengths ranging from 780 to 900 nm.

### 2.3. HT-29 Xenograft Mouse Model

Animal care, experiments, and euthanasia were performed in accordance with protocols approved by the Chonnam National University Animal Research Committee (CNU IACUC-H-2017-64). Adult (6-week-old) male NCRNU nude mice weighing approximately 25 g (N = 3 independent experiments) were purchased from Orient Bio Inc. (Seongnam, South Korea). HT-29 human colorectal adenocarcinoma cell line was obtained from the American Type Culture Collection (ATCC^®^ HTB-38^™^). Cancer cells (1 × 10^6^ cells per mouse) were harvested and suspended in 100 μL of PBS followed by subcutaneous injection into the right flank of each mouse. When the tumor size reached ~1 cm in diameter, PBS, free ICG, and the ICG-CD complex were administered intravenously. Animals were sacrificed and imaged over a certain period of time.

### 2.4. In Vivo Tumor Imaging

In vivo NIR fluorescence imaging was performed using a FOBI fluorescence imaging system (NeoScience, Suwon, South Korea). Tumor fluorescence intensity was calculated using ImageJ version 1.45q. All NIR fluorescence images were normalized identically for all conditions. To confirm the in vivo antitumor effect, the macroscopic features of each group were observed at fixed time intervals for a week. The tumor volumes were calculated using the following formula: V = 0.5 × longest diameter × (shortest diameter)^2^.

### 2.5. Assessment of In Vivo Photothermal Effect

HT-29 tumor-bearing mice were intravenously administered with PBS, free ICG, and the ICG-CD complex. At 4 h after injection, the mice were anesthetized, and their tumor sites were laser-irradiated (1.1 W/cm^2^, λ = 808 nm) for 5 min. Temperature changes in the tumor sites were monitored using a FLIR^®^ thermal imager (FLIR Systems, Wilsonville, OR, USA) and data were recorded from the beginning of the laser irradiation at a step-size of 1 min during the entire laser irradiation period. At 24 h post-irradiation, tumors were excised from the treated mice for subsequent histological analysis using hematoxylin and eosin (H&E) staining.

### 2.6. Statistical Analysis

Statistical analysis was carried out using a one-way ANOVA followed by Tukey’s multiple comparisons test. Differences were considered statistically significant at a level of *p* < 0.05. Results are presented as the mean ± S.D. and curve fitting was performed using Prism version 4.0a software (GraphPad, San Diego, CA, USA).

### 2.7. Histological Analysis

Resected tumors were preserved for H&E staining and microscopy assessment. Samples were fixed in 2% paraformaldehyde and flash frozen in optimal cutting temperature (OCT) compound using liquid nitrogen. Frozen samples were cryosectioned (10 µm in thickness per slide), stained with H&E, and observed by microscopy. Histological imaging was performed using a Nikon Eclipse Ti-U inverted microscope system (Nikon, Seoul, South Korea). Image acquisition and analysis were performed using NIS-Elements Basic Research software (Nikon).

## 3. Results and Discussion

### 3.1. Preparation and Characterization of the ICG-CD Complex

Since ICG is a conventional amphiphilic tricarbocyanine dye owing to the presence of hydrophilic sulfonyl and hydrophobic naphthyl groups in its chemical structure, it can form dimers and oligomers in aqueous solutions. It is known that the ICG aggregates cause self-quenching, which reduces fluorescence intensity in aqueous solutions [15]. To overcome the low water stability and tumor targetability of ICG, an ICG-CD complex was designed to improve the in vivo bioavailability of ICG against tumors for application in targeted photothermal cancer treatment. The preparation of the ICG-CD complex is schematically described in Figure 1. Since inclusion complexes comprising Mβ-CD have many advantages, such as improved water solubility, stability and bioavailability, we expected that the ICG-CD complex could have high optical and chemical stability and consequently achieve high tumor accumulation for enhanced PTT. As shown in Figure 2a, the absorption peaks of the ICG-CD complex measured in PBS were similar to those of ICG between the wavelengths of 710 nm and 780 nm under the same condition. Importantly, the absorbance of the ICG-CD complex at 710 nm (known as dimer λ_max_) reduced, while the intensity of the absorption peak of the ICG-CD complex at 780 nm (known as monomer λ_max_) significantly increased compared with the same concentration of free ICG. This indicates that Mβ-CD could play a role in preventing the dimeric aggregation of ICG after forming inclusion complexes with free ICG. Moreover, the ICG-CD complex showed significantly higher fluorescence intensity than free ICG under the same condition (Figure 2b). These results demonstrate that the ICG-CD complex was successfully prepared using a host–guest self-assembly process in aqueous solution and the Mβ-CD could help in improving the stability and optical properties of ICG in water.

### 3.2. Assessment of In Vitro Photothermal Effect

Efficient light-to-heat conversion is an important prerequisite for the successful therapeutic efficacy of a photothermal agent. The ICG-CD complex solution (100 μM, equivalent to a 0.3 mg/kg single dose of free ICG), free ICG, and PBS were irradiated by an 808 nm NIR laser (1.1 W/cm^2^) for 1 min and the temperature changes were continuously recorded using a FLIR^®^ thermal imager. As expected, the temperature change of the ICG-CD complex was similar to that of the free ICG solution. The temperature was immediately elevated from ambient temperature (25.6 °C) to 89.1 °C for the ICG-CD complex solution, and only a minor change was observed for PBS after 1 min of irradiation (Figure 3a). Moreover, the ICG-CD complex solution showed a rapid elevation in temperature to ~80 °C within the first 30 s, and the temperature curve attained a plateau within 1 min of laser irradiation (Figure 3b). This indicated that the ICG-CD complex could be an effective PTT agent for the photothermal cancer treatment. In addition, the absorbance of the ICG-CD complex at 780 nm gradually reduced upon prolonged irradiation with the 808 nm laser (1.1 W/cm^2^) for 5 min, indicating that ICG was photobleached after the completion of light-to-heat conversion performance (Figure 3c).

### 3.3. In Vivo NIR Fluorescence Imaging for Tumor Targetability

Real-time NIR fluorescence imaging was performed to identify the in vivo tumor targetability and biodistribution behavior of free ICG and the ICG-CD complex in HT-29 xenograft tumor models. To investigate the accumulation of the ICG-CD complex at the tumor sites compared with free ICG, the tumor-bearing mice were intravenously injected with a single dose of free ICG or the ICG-CD complex (10 nmol, 0.3 mg/kg calculated using the ICG concentration) and imaged at different time points, as shown in Figure 4a. Results from the time-dependent NIR fluorescence imaging showed that the fluorescence intensity at the tumor site upon treatment with the ICG-CD complex reached a maximum at 4 h post-injection, while the fluorescence signal from the tumor injected with free ICG showed a continuous decrease without peak accumulation until 24 h after administration (Figure 4b). These results indicate that the Mβ-CD moieties of the ICG-CD complex play an important role in improved tumor targeting such that the ICG-CD complex can be used for enhanced photothermal cancer treatment. Additionally, we confirmed the biodistribution of free ICG and the ICG-CD complex by monitoring fluorescence signals in major organs collected from mice at 4 h post-injection (Figure 4c,d). Expectedly, free ICG uptake was detected mainly in the liver, duodenum, and intestinal tract due to hepatobiliary clearance, which is a well-known characteristic of ICG. In comparison, ICG-CD complex uptake was similar to that of free ICG in the duodenum and intestinal tract, with the exception of the liver. Importantly, the differential uptake in the liver between free ICG and the ICG-CD complex might be a consequence of the reticuloendothelial system-stealthy feature of Mβ-CD. Since Mβ-CD could help in improving the water solubility and stability of ICG, the ICG-CD complex could escape from the liver and achieve prolonged blood circulation resulting in preferential tumor uptake.

### 3.4. Assessment of In Vivo Photothermal Effect

After confirming the photothermal effect of the ICG-CD complex in vitro, the PTT capability of the complex in vivo was investigated in the HT-29 xenograft tumor model. A single dose of the ICG-CD complex (10 nmol based on the ICG concentration) was intravenously injected into tumor-bearing mice at 4 h prior to 808 nm laser irradiation (1.1 W/cm^2^ for 5 min). In addition, PBS and free ICG were separately tested under the same conditions to verify the photothermal effect generated by only laser power. The temperature variations at the tumor sites in each group were minutely recorded using a FLIR^®^ thermal imager. Under laser irradiation, a rapid increase in tumor temperature to ~56.2 °C was detected in the ICG-CD complex-treated group, while only small increases in tumor temperature to ~40.3 °C and ~45.1 °C were observed in the PBS- or free ICG-injected groups, respectively (Figure 5a). These results suggest that the marked change in temperature on the tumor tissue was mediated by the improved tumor targetability of the ICG-CD complex. The tumor temperature in ICG-CD complex-treated mice rapidly increased to ~51 °C within 2 min of laser irradiation, which was sufficient to induce complete tumor necrosis; in contrast, the tumor temperatures in other groups were maintained in the range of 40–45 °C until 5 min, which was insufficient for tumor suppression (Figure 5b).

### 3.5. In Vivo PTT Efficacy

Based on its excellent in vitro photothermal effect and improved tumor accumulation in vivo, the ICG-CD complex was found to be suitable for use as a highly efficient phototherapeutic agent for fluorescence-guided PTT in tumor-bearing mice. Tumor growth in HT-29 xenograft tumor models was continuously monitored for 9 days after PTT to determine the antitumor effect of the ICG-CD complex (Figure 6a). Representative images of mice from each group were acquired to reflect the change in tumor size. While the tumor volumes of the groups treated with PBS and free ICG before laser irradiation increased three to four-fold during the observation period of 9 days, tumor volumes in the ICG-CD complex-treated group rapidly decreased within 5 days of laser irradiation without tumor regrowth (Figure 6b). Due to the improved accumulation of the ICG-CD complex at the tumor sites, the combination of the ICG-CD complex and laser irradiation induced tumor ablation and only black scars were observed at the tumor site after 3 days of treatment. Moreover, no body weight loss was observed in the tumor mice during the course of the therapy, indicating that this treatment was safe and harmless (Figure 6c). Further evaluation was conducted by H&E staining of tumor tissue samples collected from each group after 1 day of treatment (Figure 6d). In the group treated with the combination of the ICG-CD complex and laser irradiation, significant cell damage, such as nuclear damage and cell shrinkage, was observed. In contrast, the tumor tissues from other groups showed vigorous cell proliferation and tight arrangements without any detectable damage. These results demonstrate that the ICG-CD complex is a promising candidate for effective cancer phototherapy.

## 4. Conclusions

In summary, a phototherapeutic ICG-CD complex was successfully prepared by host–guest inclusion complexation of ICG and Mβ-CD in aqueous state, which was further used to improve its tumor accumulation for enhanced photothermal performance. In the present study, the ICG-CD complex exhibited higher performance in targeted tumor imaging than free ICG. Mβ-CD promoted the tumor targetability of ICG, further increasing its PTT efficacy. In particular, fluorescence-guided cancer PTT suggested a highly effective outcome, leading to apparent tumor ablation without damaging adjacent normal tissues. Therefore, the ICG-CD complex could be a promising phototherapeutic agent for targeted tumor imaging and precise cancer PTT in future clinical applications.

## Figures and Tables

**Figure 1 biomedicines-08-00476-f001:**
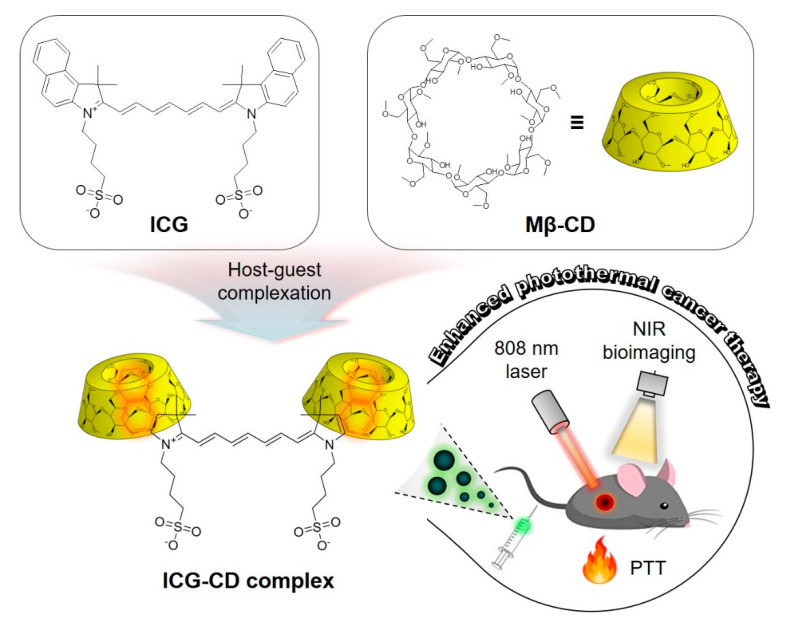
Schematic representation of inclusion complex formation between indocyanine green (ICG) and methyl-β-cyclodextrin (Mβ-CD) in an aqueous state for enhanced photothermal cancer therapy. PTT: photothermal therapy.

**Figure 2 biomedicines-08-00476-f002:**
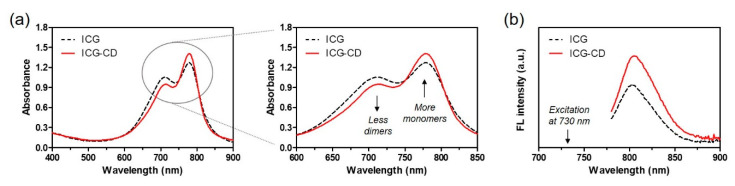
Optical properties of free ICG and the ICG-CD complex. (**a**) Changes in absorbance spectra of the ICG and ICG-CD solutions at the same ICG concentration. Amplification of the 710 nm and 780 nm regions showing changes in absorbance of ICG-dimeric and ICG-monomeric forms in PBS. (**b**) Fluorescence spectra of ICG and ICG-CD solutions upon excitation at 730 nm. Absorbance and fluorescence emission spectra of each sample were obtained at a fixed ICG concentration of 10 μM in PBS at pH 7.4.

**Figure 3 biomedicines-08-00476-f003:**
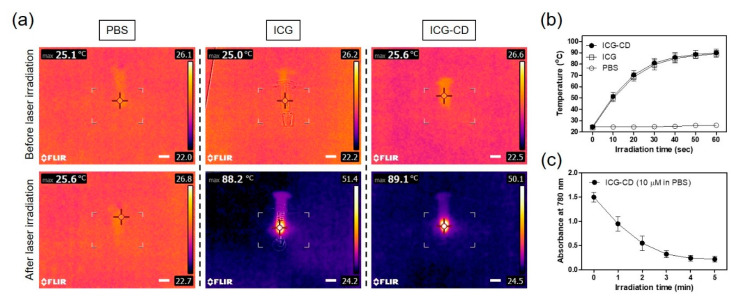
(**a**) In vitro photothermal images of the free ICG and ICG-CD complex diluted in PBS (10 nmol based on the absorbance of ICG; 100 μM is equivalent to a single dose of 0.3 mg/kg of ICG) and PBS alone (100 μL) exposed to 808 nm laser (1.1 W/cm^2^) for 1 min. The maximum temperature was automatically recorded using an infrared thermal camera as a function of irradiation time. Scale bars = 1 cm. (**b**) Temperature changes in the solutions in each sample were monitored during 1 min of laser irradiation. (**c**) Changes in absorbance of the ICG-CD complex solution (10 μM) were measured at 780 nm during 5 min of laser irradiation. Data are expressed as the mean ± S.D. of three independent experiments.

**Figure 4 biomedicines-08-00476-f004:**
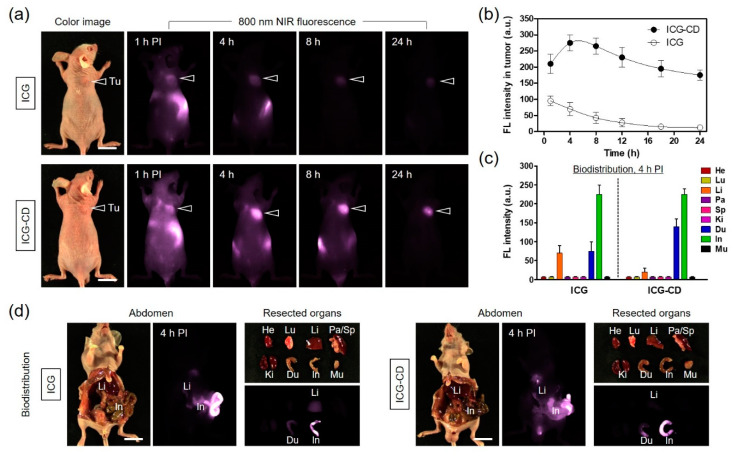
In vivo HT-29 tumor targeting efficiency of free ICG and the ICG-CD complex. (**a**) NIR fluorescence imaging for 24 h post-injection of ICG and the ICG-CD complex. (**b**) Time-dependent fluorescence intensity at tumor sites targeted by ICG and the ICG-CD complex. (**c**) Quantitative fluorescence analysis of intraoperative dissected organs at 4 h post-injection of ICG and the ICG-CD complex. (**d**) Abdominal exploration and resected organs imaged at 4 h post-injection of ICG and the ICG-CD complex. Tumor mice were intravenously injected with 10 nmol of ICG or the ICG-CD complex and imaged for 24 h. The tumor site is indicated by an arrowhead. Abbreviations: Du, duodenum; He, heart; In, intestine; Ki, kidneys; Li, liver; Lu, lungs; Mu, muscle; Pa, pancreas; Sp, spleen; Tu, tumor; and PI, post-injection. Scale bars = 1 cm. Images are representative of three independent experiments. All NIR fluorescence images have identical exposure and normalization. Data are expressed as the mean ± S.D. of the three independent experiments.

**Figure 5 biomedicines-08-00476-f005:**
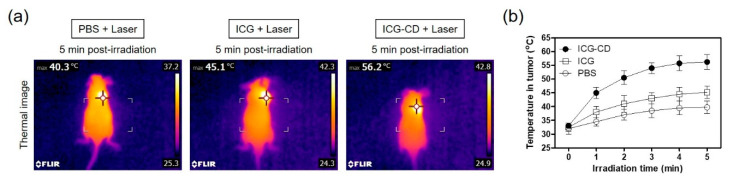
(**a**) Whole-body thermal images of tumor-bearing mice at 4 h post-injection of PBS, ICG, or the ICG-CD complex upon exposure to 808 nm laser irradiation (1.1 W/cm^2^) for 5 min. The maximum tumor temperatures were automatically recorded with an infrared thermal camera as a function of irradiation time. (**b**) Temperature changes at the tumor sites in each treatment group were monitored during 5 min of 808 nm laser irradiation. Data are expressed as the mean ± S.D. of three independent experiments.

**Figure 6 biomedicines-08-00476-f006:**
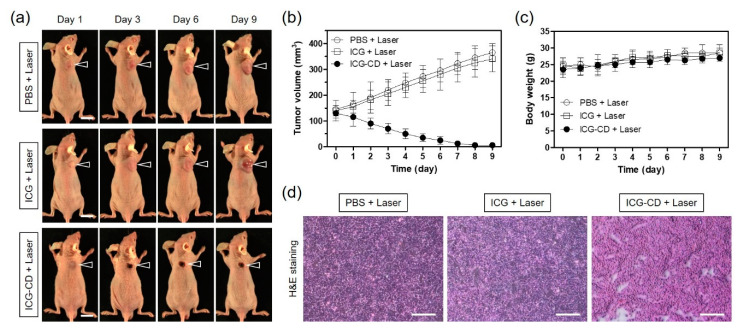
(**a**) In vivo NIR phototherapeutic efficacy. Representative images showing changes in tumor size with 4 h post-injection of PBS, ICG, and the ICG-CD complex, followed by 808 nm laser irradiation (1.1 W/cm^2^) for 5 min. The tumor site is indicated by an arrowhead. Scale bars = 1 cm. (**b**) Tumor growth rate and (**c**) body weight of each treatment group were monitored for 9 days. Data are expressed as the mean ± S.D. of three independent experiments. (**d**) Tumor sections from each group stained with H&E at 24 h after laser irradiation. Scale bars = 100 μm.

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
