# Peer review of "Indocyanine Green and Methyl-β-Cyclodextrin Complex for Enhanced Photothermal Cancer Therapy"

_biomedicines, 2020, doi:10.3390/biomedicines8110476_

Round 1

Reviewer 1 Report

The authors have successfully demonstrated using CD as a carrier of ICG to improve the delivery and, thus, the therapeutic efficacy of photothermal therapy. It will be interesting to see if the same approach can be used to treat cancers using other animal models.

Minor

(1) Combine paragraph one and two

(2) The author may consider include additional characterizations of the ICG-CD complex (to confirm the formation of complex) rather than just the optical properties.

(3) In Figure 3, the author may consider comparing the light-induced temperature changes between free ICT and ICT-CD complex.

Author Response

The authors have successfully demonstrated using CD as a carrier of ICG to improve the delivery and, thus, the therapeutic efficacy of photothermal therapy. It will be interesting to see if the same approach can be used to treat cancers using other animal models.

Thank you very much for your kind comments. We are planning to test the theranostics in other tumor xenograft models (NCI-H460, MCF-7, and HepG2) with the same approach for the next study.

(1) Combine paragraph one and two

In response to the Reviewer’s suggestion, we combined the paragraphs 1 and 2 in the Introduction part.

(2) The author may consider include additional characterizations of the ICG-CD complex (to confirm the formation of complex) rather than just the optical properties.

This is an important point. Although the technical methods for confirming the inclusion complexation between ICG and CD may be varied including NMR spectroscopy, the optical analysis is a popular class of methods to prove the formation of complex, as reported previously (cited in References #14 and #21 (Nano Res. 2020, 13, 1100–1110.; J. Phys. Org. Chem. 2010, 23, 893–903.). We thank the Reviewer for kind understanding on this matter.

(3) In Figure 3, the author may consider comparing the light-induced temperature changes between free ICT and ICT-CD complex.

We agree to the Reviewer’s comment and tested the temperature change of free ICG solution in Figures 3a and b. Also, we added the explanation in the Results and discussion part as “As expected, the temperature change of the ICG-CD complex was similar to that of free ICG solution.”.

Reviewer 2 Report

This manuscript demonstrates an effective method to improve the pharmacokinetic property of indocyanine green (ICG) for enhanced tumor delivery and photothermal therapy by preparing a complex with methyl-beta-cyclodextrin (MbCD). I was particularly intrigued by how such a simple approach (i.e., complex with MbCD) had resulted in a remarkable tumor regression by photothermal therapy. The studies have been well designed to support their hypothesis and the results are scientifically sound. I recommend this manuscript to be accepted and proceeded further for the publication in the Biomedicines. One thing I would like to suggest to the authors is to make the scale consistent throughout the images in Figure 3a, so that the figure could be more readily understandable to the readers of the article.

Author Response

Thank you very much for your kind comments. We are very pleased to publish our work in this journal. In response to the Reviewer’s suggestion, we added the scale bars in the thermal images, as shown in Figure 3a.

Reviewer 3 Report

The manuscript is focused on the development of ICG-CD complex for tumor imaging and photothermal therapy. The development of efficient theranostic approaches for selective tumor imaging and effective photothermal therapy is still of great need.

I find the experimental data are largely sound and the authors cover the in vitro and in vivo tests, so according to my opinion this article can be accepted for publishing in Biomedicines.

Author Response

Thank you very much for your kind comments. We are very pleased to publish our work in this journal.